# Super-Resolution Microscopy Reveals a Direct Interaction of Intracellular *Mycobacterium tuberculosis* with the Antimicrobial Peptide LL-37

**DOI:** 10.3390/ijms21186741

**Published:** 2020-09-14

**Authors:** Dhruva Deshpande, Mark Grieshober, Fanny Wondany, Fabian Gerbl, Reiner Noschka, Jens Michaelis, Steffen Stenger

**Affiliations:** 1Institute of Biophysics, Ulm University, 89081 Ulm, Germany; deshpandedhruva@gmail.com (D.D.); fanny.weiss@uni-ulm.de (F.W.); 2Institute of Medical Microbiology and Hygiene, University Hospital Ulm, 89081 Ulm, Germany; mark.grieshober@uniklinik-ulm.de (M.G.); fabian.gerbl@uni-ulm.de (F.G.); reiner-martin.noschka@uni-ulm.de (R.N.)

**Keywords:** LL-37, antimicrobial peptide, *Mycobacterium tuberculosis*, STED, human macrophages

## Abstract

The antimicrobial peptide LL-37 inhibits the growth of the major human pathogen *Mycobacterium tuberculosis* (*Mtb*), but the mechanism of the peptide–pathogen interaction inside human macrophages remains unclear. Super-resolution imaging techniques provide a novel opportunity to visualize these interactions on a molecular level. Here, we adapt the super-resolution technique of stimulated emission depletion (STED) microscopy to study the uptake, intracellular localization and interaction of LL-37 with macrophages and virulent *Mtb*. We demonstrate that LL-37 is internalized by both uninfected and *Mtb* infected primary human macrophages. The peptide localizes in the membrane of early endosomes and lysosomes, the compartment in which mycobacteria reside. Functionally, LL-37 disrupts the cell wall of intra- and extracellular *Mtb*, resulting in the killing of the pathogen. In conclusion, we introduce STED microscopy as an innovative and informative tool for studying host–pathogen–peptide interactions, clearly extending the possibilities of conventional confocal microscopy.

## 1. Introduction

Despite the availability of effective antibiotic treatment of tuberculosis, the management of this devastating disease remains a major challenge. The increasing frequency of drug-resistant strains, as well as severe side effects along with lack of compliance associated with the long duration of treatment, calls for the development of alternative treatment strategies [1,2]. One intriguing possibility is the use of endogenous antimicrobial peptides (AMP), which form an integral part of the human innate immunity system. AMPs are small peptides, which show broad range of activity against human pathogens [3]. Cathelicidin is a cationic antimicrobial peptide (hCAP18) that is up-regulated in human macrophages by a vitamin D-dependent pathway [4]. It is cleaved into the active form LL-37, which is active against extra- and intracellular *Mycobacterium tuberculosis* (*Mtb*). The peptide is highly expressed in patients suffering from active pulmonary tuberculosis [5,6]. Several hypothesis have been proposed to explain the mechanism by which AMPs interact with *Mtb*, eventually causing cell death [7]. One major mechanism involves the disruption of the bacterial cell wall, eventually causing fatal cytoplasmic leakage. Other possibilities include the impairment of cellular processes and pore formation caused by ionic interactions of the peptide and the bacterial cell wall [3,8,9].

Mycobacteria have a complex cell envelope with limited permeability, thereby conferring resistance against conventional antimicrobial compounds [10,11]. The outer layer, called the mycomembrane, is mainly composed of proteins, glycolipids and lipoglycans [12,13,14]. Lipoarabinomannan (LAM) is a glycolipid abundantly expressed in the cell envelope and plays an important role in modulating innate and adaptive host immune response [15,16,17,18]. Inhaled *Mtb* are phagocytosed by macrophages and enclosed in intracellular vesicles [19]. *Mtb* can prevent the formation of acidic phagolysosomes and the bacteria sequestered in the phagosomes are protected from degradation. This occurs by inhibiting the recruitment of the early endosomal antigen 1 protein (EEA1) to the phagosomal membrane. EEA1 contributes to the maturation of phagosome to lysosomes [15,20,21].

For internalization of LL-37 into macrophages, clathrin-mediated or lipid-raft dependent endocytosis have been suggested [22,23,24]. In infected macrophages, LL-37 enters phagosomes where intracellular bacteria are either directly eliminated by bacterial cell wall degradation or indirectly by inducing the cellular process of degradation with lysosomal fusion [10]. Hence, there are several different models under debate for the interaction of LL-37, macrophages and *Mtb*.

A variety of methods have been used to study *Mtb*, LL-37 and their interaction with macrophages, including biochemistry [25,26], molecular biology [27] and modeling [28]. Several imaging approaches have also been developed, each having specific advantages [29,30,31]. In particular, fluorescence microscopy, which employs specific fluorescent labelling of cellular and molecular targets to co-label different proteins and to localize them to specific cellular compartments. However, diffraction limits the achievable resolution for optical wavelengths to about ~250 nm laterally (in the x-y plane) and ~600 nm in the axial direction (along the optical z-direction). This makes it difficult to visualize defined structures or distinguish between two distinct labelled points positioned closer than the diffraction limit with conventional optical microscopy. Electron microscopy allows sub-nanometer resolution, but has several drawbacks such as the limited number of targets that can be labelled simultaneously [32]. A solution to this problem is provided by the development of super-resolution (SR) optical imaging techniques, which can bypass the diffraction-limit, offering a (close to) molecular level resolution, while maintaining the benefits of the established immunolabelling approach of fluorescent microscopy. Several different SR approaches have been developed including the confocal-based stimulated-emission depletion (STED) microscopy, single-molecule localization microscopy (SMLM), and saturated-structured illumination microscopy (SSIM) [33,34,35,36,37]. Like in confocal microscopy, STED utilizes a focused laser beam, which is scanned across the sample to excite fluorescence. However, the diffraction limit of light is subverted by superimposing the excitation with an additional donut-shaped depletion beam, with an intensity minimum in the center (donut-hole). Thus, the fluorescence signal is spatially confined by stimulated emission depletion. While theoretically, the resolution in STED is not limited, practically considering the phototoxicity tolerance of the biological sample and the laser power available, an axial resolution of ~35 nm and laterally of ~90 nm can be achieved [38]. With the advent of easier sample preparation and the availability of commercial STED microscopes, STED microscopy is an emerging option for investigating questions in biomedical research, including host–pathogen and AMP–pathogen interactions [39,40,41]. In this study, we evaluated the suitability of innovative STED microscopy as a tool for directly observing host–pathogen interaction using, as an example, LL-37, primary human macrophages and virulent *Mtb*.

## 2. Results

### 2.1. Detection of Lipoarabinomannan by STED Microscopy and Confocal Microscopy

Lipoarabinomannan (LAM) is a major component of the mycobacterial cell wall and a potential target for antimicrobial peptides. To detect LAM and visualize the mycobacterial cell wall, we compared confocal- and STED microscopy (Figure 1A). In the confocal microscopy images, it is not possible to define the spatial localization of LAM within the bacterium. In contrast, in STED imaging performed with the same instrument as used for confocal microscopy, the identical bacterium shows a distinct labelling of the cell surface but very little labelling of the cytoplasm. This can also be quantified by taking a cross-section across the bacterium (Appendix A). While in the STED image, the maximum of the intensity is at the cell periphery, in confocal imaging, it appears as if the brightest spot is at the cell interior. In addition, a scan along the axial direction (xz-scan) on the identical bacterium shows an intensity minimum at the cell interior in STED microscopy, while a maximum is observed in confocal microscopy (Figure 1B). Thus, the superior resolution of STED imaging allows a more precise interpretation of the immunofluorescent labelling and definition of the localization of the target structure (LAM).

To investigate whether STED microscopy is also suitable for analyzing the interaction between *Mtb* and antimicrobial peptides, we labelled extracellular *Mtb* with an Atto647N conjugated NHS-ester, which non-specifically binds to amino-bonds. Labelled bacilli were exposed to LL-37 peptide, which is directly conjugated to a fluorescent dye (TAMRA) for 5 min. Sections acquired with confocal microscopy showed bacterial cells with a spatially overlapping signal in both channels (Figure 1C). In the STED image, the LL-37 signal, which for this imaging condition appears patchy, is clearly distinguishable from the labelling of the cell wall. Note that for this image, in contrast to the image shown in Figure 1a, the focus was adjusted not to the cell interior but to the cell surface, such that labelling is visible throughout. This example illustrates a major advantage of STED over confocal microscopy. While a common pitfall of confocal laser microscopy is an artificial overlap of closely localized molecules, STED microscopy offers a superior spatial resolution range and therefore allows for clear distinction of most cellular components. 

### 2.2. Effect of LL-37 on Extracellular Mtb Visualized by STED Microscopy

Prior to investigating the effect of LL-37 on the morphology of the mycobacterial cell wall by STED microscopy, we ascertained that TAMRA-labelling (required for immune-detection of LL-37) does not affect the antimicrobial activity. By measuring the incorporation of ^3^H-labelled uracil by LL-37 TAMRA (11 µM)-treated *Mtb*, we show that LL-37-TAMRA retains antimicrobial activity (Figure 2A). Next, we visualized the effect of LL-37 on *Mtb* after 5 and 30 min of incubation by STED microscopy. To understand the mechanism of action, we had to reduce the LL-37-TAMRA concentration, since the active concentration was too bright to apply to STED microscopy. After 5 min, LL-37 localized to the cell periphery whereas the cytoplasm remained un-stained (Figure 2B), indicating a fairly homogenous distribution of the peptide to the bacterial cell wall. As for the case of LAM labelling, the interior of the cell showed negligible intensity when imaged using STED microscopy both in the lateral as well as in the axial direction. After 30 min, the cell wall appeared disrupted with few peptide clusters discernible. Additionally, LL-37 clusters are visible in the vicinity of the bacterial cell, suggesting rupture of the cell wall. After extension of the incubation period to 60 min, no rod-shaped bacteria were detectable anymore, indicating that cells had completely disintegrated. Taken together, our results demonstrate that LL-37 rapidly localizes to the mycobacterial cell wall, subsequently causing disruption of the bacilli. Interestingly, while it was easily possible to label the cell wall of untreated bacteria with anti-LAM antibody, similar labelling could not be visualized in cells treated with LL-37. There was no indication that LL-37 is internalized into the mycobacterial cytoplasm, supporting the concept that rupture is directly or indirectly triggered by the interaction of LL-37 molecules with the bacterial cell wall.

### 2.3. LL-37 Is Taken Up by Macrophages and Colocalizes with Early Endosomes

Since *Mtb* is an intracellular pathogen residing primarily within macrophages, we investigated whether LL-37 is taken up by the mycobacterial host cells. Macrophages were incubated with LL-37-TAMRA and the uptake and intracellular localization was determined after 30 min by confocal- and STED microscopy. Both imaging techniques clearly revealed an efficient uptake of LL-37 in macrophages (Figure 3). Uptake of LL-37 is likely an active process rather than passive diffusion because a significant reduction in intracellular peptide was observed when cultures were incubated at 4 °C, as determined by flow cytometry (Appendix A). Confocal images demonstrate punctate but homogenous labelling within macrophages, indicating that LL-37 is internalized (Figure 3A). STED microscopy not only revealed the uptake of LL-37, but also distinct clusters of LL-37 in donut-like formations (Figure 3B), again demonstrating the superiority of STED microscopy to identify distinct subcellular structures. 

The donut-shaped staining pattern suggested that LL-37 accumulates in the membrane of cellular vesicles, e.g., endosomes of different maturation stages and/or lysosomes. To test this hypothesis, we labelled early endosomes of LL-37-treated macrophages with EEA1 antibodies. Dual color STED microscopy images showed a striking colocalization of the early endosomal marker (EEA1) with LL-37 (Figure 3B). Virtually all detectable LL-37 was located in the endosomal membrane. Accordingly, co-labelling of LL-37-TAMRA with lysosomal-associated membrane protein 1 (LAMP1) showed no significant colocalization (Appendix A). Both the EEA1 and LL-37 form donut shape structures, indicating that both are localized in the endosomal membrane. The donut-like shape is unlikely to be an artifact, since molecules that are located inside endosomal vesicles appear as punctate objects inside the lumen of the endosomes in STED images [42]. To check the successful internalization of *Mtb* in macrophages under our experimental conditions, we performed a LAM-staining on intracellular *Mtb* and a microtubule staining for the macrophages (Appendix A). 

### 2.4. LL-37 Localizes in Endosomes and Lysosomes of Infected Macrophages

To exert antimicrobial activity, LL-37 is likely not only taken up by the microbial host cell, but also co-localizes with the intracellular pathogen. We tested this possibility by incubating *Mtb*-infected macrophages with LL-37-TAMRA overnight and staining for early (EEA-1)- or late (LAMP-1)- endosomes. As already demonstrated for uninfected cells, LL-37 also co-localized with early endosomes in *Mtb*-infected macrophages. (Figure 4A). In addition, LL-37 co-localized with LAMP-1-positive vesicles (Figure 4B), which was not observed in un-infected macrophages. In infected, but not in un-infected macrophages, LL-37 labelled vesicles showed an elongated morphology (Figure 4A, arrowhead). The shape of the structures resembled the mycobacterial rod-like shape. Quantitative analysis of the perimeter confirmed that these vesicles are elongated as compared to the LL-37-positive vesicles in un-infected macrophages (Figure 4C), possibly reflecting association with intracellular *Mtb*. In control samples infected with *Mtb* but not exposed to LL-37, we did not observe the formation of a phagolysosome-like structure labelled with EEA1 or LAMP-1 (Appendix A).

### 2.5. Antimicrobial Activity of LL-37 against Intracellular Mtb

These findings indicated an interaction between *Mtb* and LL-37 in macrophages. To assess the functional implication of this observation, we infected primary human macrophages with virulent *Mtb* and treated them with (unlabelled) LL-37. After 4 days, the viability of intracellular bacteria was determined by plating cell lysates and determining the number of CFU. LL-37 decreased the viability of *Mtb* in all six donors investigated (Figure 5A), confirming previous results by us and others [4,43,44]. To investigate whether the antimicrobial effect of LL-37 correlated with morphological alterations of the bacteria, we employed the STED microscopy technique we had previously established for extracellular bacilli (Figure 1). The images demonstrate that LL-37 co-localizes with the intracellular mycobacteria (detected by LAM-labelling) and show severe fragmentation of the bacterial cell wall (Figure 5B). 

In summary, our results establish STED super-resolution microscopy as a powerful tool to detect and characterize the interaction of antimicrobial peptides and intracellular pathogens. Refinement of STED microscopy by including additional wavelengths and live cell imaging will provide a unique opportunity to understand the molecular mechanism and lay the basis for the functional optimization of antimicrobial peptides.

## 3. Discussion

The major aim of this study was to establish super-resolution optical microscopy as a novel tool to gain mechanistic insight into the interaction of bacterial pathogens and antimicrobial peptides. Using a relevant model of human macrophages, virulent *Mtb* and the well characterized antimicrobial peptide LL-37, our results demonstrate the advantage of using STED microscopy for imaging intracellular structures as compared to confocal imaging. The nanometer-level resolution allowed us to distinguish between membrane embedded and internalized molecules in cell organelles. Specifically, we established the membrane association of LL-37 in bacteria and endosomes. When analyzing confocal microscopy images, “yellow” overlap in dual colored images (with red and green channels) is often misinterpreted as colocalization. When the same region is visualized with STED microscopy, more details are resolved and colocalization observed by confocal imaging is often proven to be misleading. The high spatial resolution of the STED technique thus demonstrated the colocalization of *Mtb* and LL-37 in macrophage compartments, greatly reducing the pitfalls of ‘false co-localization’ seen in diffraction-limited optical microcopy. Therefore, we provide evidence that STED microscopy is a feasible technique for biological imaging studies that by far exceeds the resolution of confocal microscopy and hence allows for sub-compartimental localization. It has, however, to be noted that the maximum resolution offered by this technique is ~35 nm, so co-localization errors can occur with distances smaller than the resolution limit.

In this study, we used STED microscopy to localize LL-37 on extracellular *Mtb*. The peptide initially located to the bacterial cell wall where it presumably interacts with the glycolipid LAM. Over time, we observed a disruption of the mycobacterial cell wall correlating with reduced viability of the bacilli. We detected a similar morphology in *Mtb*-infected macrophages, where LL-37 is internalized, colocalizes with *Mtb* in vesicular organelles and induces partial killing of the bacilli. Several studies support such a direct activity of antimicrobial peptides against *Mtb* [45,46].

For these experiments, a higher multiplicity of infection (MOI) was necessary to reduce the time to locate infected macrophages by STED microscopy. Prolonged examination of the slides results in fading of the fluorescence. In addition, the incubation times had to be adjusted according to the objective of the experiments. For example, LL-37 shows a reduction in intracellular bacteria after 72 h (Figure 5A). Therefore, earlier time points had to be investigated to understand the uptake mechanism and the LL-37–macrophages interaction, which occurs prior to killing (Figure 3). For example, a disruption of the *Mtb* cell wall when treated with LL-37 and other synthetic peptides has been reported [47]. Peptides with an α-helical structure, like LL-37, often show a flexible configuration in solution but fold into amphipathic, often alpha-helical, structures upon contact with membranes, allowing their insertion into the hydrophobic interior of the bilayer [3,27]. However, whether the deleterious effect is mediated by permeabilization or pore-formation is a means to reach an internal target, e.g., DNA, remains unclear. Pore-formation in model lipid bilayer membrane by lipid–protein interactions has been observed previously using STED microscopy [48]. Our results suggest that peptides bind to the bacterial cell wall, causing dis-integration and rapid rupture within minutes. 

Labelling of LAM in LL-37-treated extracellular *Mtb* was not feasible (data not shown), indicating a possible interaction of the peptide with LAM, as proposed previously [44]. Further support for this hypothesis comes from our observation that LL-37 induces cell wall rupture in intracellular *Mtb* detected by LAM (Figure 5). The glycolipid LAM has been proposed to prevent the fusion of phagosomes with lysosomes [16]. Further, stimulation of LAM triggers the production of LL-37 [10,47]. It is tempting to speculate that the targeted disruption of the virulence factor LAM is a potential mechanism by which LL-37 exerts antimicrobial activity.

Our results showed that LL-37 is internalized by macrophages and localizes within the membrane of early endosomal labelled with EEA1. In *Mtb*-infected cells, a proportion of LL-37 and EEA1 co-labelled vesicles showed an elongated morphology with increased size. These structures likely correspond to phagosomes, since, apart from their larger size and shape, phagosomes share similar membrane morphology and luminal composition with an early endosome and hence are also labelled with the anti-EEA1 antibody [20,49]. Phagosomes are cytoplasmic vesicles which internalize foreign particles and fuse to form phagolysosomes. Ingested particles are exposed to the hostile acidic environment and are eventually degraded by autophagy [43,50]. This defense mechanism is normally suppressed by almost all virulent *Mtb* [51], but this inhibition may be counteracted by LL-37.

The phagosome maturation process requires the recruitment of several membrane proteins. One of the early steps involves recruitment of EEA1 to the phagosome membrane either by fusion with endosome vesicles or directly from the cytosol. In the phagosomal membrane, EEA1 acts as a regulator of vesicular protein trafficking and the presence of EEA1 is vital to trigger fusion with lysosomes [21]. However, the mycobacterial phagosomes exclude EEA1 protein and this is one of the contributing factors in autophagy evasion [52]. The glycosylated form of LAM, termed mannose LAM, is one of the key *Mtb*-derived molecules interfering with EEA1 recruitment. Mannose LAM inhibits phosphadtidylinositol-3-kinase (PI3K), which is essential for mediating EEA1 acquisition to the phagosomes [15]. However, the presence of EEA1 observed in the STED images reported here, points to an efficient blocking of LAM by LL-37 in the mycobacterial phagosome membrane of the LL-37-treated macrophages. This could point to the initiation of autophagy in the presence of LL-37; whether autophagy is also stimulated by the presence of LL-37, as proposed earlier [10], cannot be tested by our experiments.

Antimicrobial peptides are promising candidates to complement conventional drug therapy in tuberculosis and show consistent activity against multi-drug resistant strains [9]. However, our knowledge on the cellular and molecular mode of action remains incomplete. Innovative imaging technologies offer exciting opportunities to address these questions from a different angle. Our results establish STED microscopy as an innovative and informative tool for studying interactions between antimicrobial peptides, host cells and the pathogen.

## 4. Materials and Methods 

### 4.1. Bacteria

*Mycobacterium tuberculosis* H37Rv (ATCC 27294, Manassas, VA, USA) was grown in suspension with constant gentle rotation in roller bottles (Corning, Corning, NY, USA) containing Middlebrook 7H9 broth (BD Biosciences, Franklin Lakes, NJ, USA) supplemented with 1% glycerol (Roth, Karlsruhe, Germany), 0.05% Tween 80 (Sigma-Aldrich, Steinheim, Germany), and 10% Middlebrook oleic acid, albumin, dextrose, and catalase enrichment (BD Biosciences, OADC) adjusted to pH 7.2–7.4. Aliquots from logarithmically growing cultures were kept frozen at −80 °C in 7H9 broth with 20% glycerol, and representative vials were thawed and enumerated for viable CFU on Middlebrook 7H11 plates (BD Biosciences, Franklin Lakes, NJ, USA). Staining of bacterial suspensions with fluorochromic substrates differentiating between live and dead bacteria (BacLight, Invitrogen, Carlsbad, CA, USA) revealed a viability of the bacteria >90% and confirmed the absence of large bacterial aggregates. Thawed bacterial stock aliquots were sonicated for 10 min at 40 kHz and 110 W in a Transsonic digital S (Elma, Wetzikon, Switzerland) before use.

### 4.2. Generation of Human Monocyte-Derived Macrophages

Human peripheral blood mononuclear cells (PBMC) were isolated by density gradient centrifugation (Ficoll-Paque Plus; GE Healthcare, Chicago, IL, USA) of buffy coat preparations from anonymous donors (Institute of Transfusion Medicine, Ulm University). Monocytes were selected from PBMCs by adherence on plastic. Monocyte-derived macrophages (referred to as “macrophages”) were generated by treatment with GM-CSF (10 ng/mL; Miltenyi Biotec, Bergisch Gladbach, Germany) in RPMI 1640 (Life Technologies, Carlsbad, CA, USA) cell culture medium supplemented with L-glutamine (2 mM; PAN Biotech, Aidenbach, Germany), HEPES (10 mM; Biochrom, Berlin, Germany), Penicillin/Streptomycin (100 µg/mL; Biochrom, Berlin, Germany) and 5% heat-inactivated human serum (Lonza, Basel, Switzerland) for 6 d. After culture, macrophages were detached with EDTA (1 mM; Sigma-Aldrich, Steinheim, Germany). Phenotypic characterization by flow cytometry demonstrated that macrophages expressed CD68 (anti–CD68-FITC, clone Y1/82A; BD Biosciences, Franklin Lakes, NJ, USA) and MHCII (anti–HLA-DR-PerCP, clone L243; BD Biosciences, Franklin Lakes, NJ, USA) as described [53].

### 4.3. Quantification of Intracellular Mycobacterial Growth

A quantity of 5 × 10^6^ Macrophages were bulk-infected in 6-well plates with single-cell suspensions of *M. tuberculosis* (multiplicity of infection (MOI = 5)) in RPMI 1640 (Life technologies, Carlsbad, CA, USA), L-glutamine (2 mM; PAN biotech, Aidenbach, Germany), HEPES (10 mM; Biochrom, Berlin, Germany), Penicillin (60 µg/mL; Biochrom, Berlin, Germany), (Amphotericin B 5.6 µg/mL; Sigma-Aldrich, Steinheim, Germany) and 10% non-heat-inactivated human serum (Lonza, Basel, Switzerland). After 2 h, macrophages were washed thoroughly to remove extracellular *Mtb* and harvested using EDTA (1 mM; Sigma-Aldrich, Steinheim, Germany). The efficacy of infection and cell viability was determined using Auramine-Rhodamine (Merck, Darmstadt, Germany) and Annexin V staining (BD, Franklin Lakes, NJ, USA), as previously described [53]. Then, 2 × 10^5^ infected macrophages were seeded in 24-well plates and incubated with LL-37 (11 µM) for 4 d. To enumerate the number of viable bacilli, infected macrophages were lysed with 0.3% saponin (Sigma-Aldrich, Steinheim, Germany). Cell lysates were vigorously resuspended, transferred in screw cap tubes and sonicated for 10 min at 40 kHz and 110 W. Afterwards, serial dilutions (1:10; 1:100, 1:1000) of the sonicates were plated on 7H11 agar plates (BD, Franklin Lakes, NJ, USA) and incubated for 14 d before counting the CFU.

### 4.4. ^3^H-Uracil Proliferation Assay

As a correlate of mycobacterial metabolism (and by inference viability), we measured the rate of RNA synthesis by *Mtb.* For this, we determined the uptake of radioactively-(^3^H)-labelled Uracil. To do this, 2 × 10^6^ sonicated *Mtb* were incubated with LL-37 (11 µM) in a 96 well plate (Thermo Fisher, Waltham, MA, USA) in Middlebrook 7H9 broth supplemented with OADC, 1% glycerol and 0.05% Tween 80. The final volume was 100 µL and all samples were set up in triplicates. After 3 days 5,6-^3^H-Uracil (ART-0282, Biotrend, 0.3 µCi, Cologne, Germany) was added and cultures were incubated for additional 18 h. *Mtb* were inactivated by adding paraformaldehyde (PFA, Sigma-Aldrich, 4% final concentration, Steinheim, Germany) for 20 min. Bacteria were transferred to glass fiber filters (Printed Filtermat A, PerkinElmer, Waltham, MA, USA) using a 96 well- cell harvester (Inotech, Nabburg, Germany). The filtermat was sealed with wax plate containing the scintillation liquid (MeltiLex, Perkin Elmer, Waltham, MA, USA) at 75 °C. Incorporation of ^3^H Uracil by the bacilli was measured using a ß-counter (Sense Beta, Hidex, Turku, Finland). The mean counts per minute of each sample were normalized to the untreated control and expressed as antimicrobial activity.

### 4.5. Treatment of Extracellular Mtb with LL-37-TAMRA

For investigating the interaction of LL-37 with *Mtb*, LL-37 conjugated to the fluorescence dye 5-TAMRA (LLGDFFRKSKEKIGKEFKRIVQRIKDFLRNLVPRTES; Innovagen, Lund, Sweden) peptide was used. Extracellular *Mtb* (2 × 10^7^) were seeded in a 12-well plate (Sarstedt, Nümbrecht, Germany) containing a high precision glass coverslip (Carl Roth, Karlsruhe, Germany; 170 µm thickness) coated with poly-L-lysine (Sigma-Aldrich, Steinheim, Germany). Afterwards, LL-37-TAMRA (22 nM) was added and incubated as indicated. Excess peptide was removed by rinsing of the wells carefully. For control experiments, bacteria (2 × 10^7^) were labelled with anti-LAM antibodies (mouse monoclonal, clone CS-35, 1:500 dilution of hybridoma supernatant) for 2 h, without LL-37 treatment. Finally, Atto647-conjugated goat anti-mouse (1:1000; Sigma-Aldrich, Steinheim, Germany) was added for 60 min. The coverslips were fixed with PFA, transferred from the well onto a microscope slide and immersed in mounting medium (2,2′-thiodiethanol; Sigma-Aldrich, Steinheim, Germany). The samples were then analysed by confocal- or STED microscopy.

### 4.6. Imaging of the LL-37 and Mtb in Macrophages

A quantity of 5 × 10^6^ Macrophages were bulk-infected with *Mtb* (MOI 50). Infected macrophages were harvested and given into a 12-well plate containing glass coverslips. LL-37-TAMRA (11 µM) was added overnight. Cells were then fixed with 4% paraformaldehyde, followed by permeabilization and blocking for 2 h in 0.3% Triton-X and 3% BSA (all Sigma-Aldrich, Steinheim, Germany). Next, *Mtb* was labelled by LAM antibodies (mouse monoclonal, clone CS-35, 1:500 dilution of hybridoma supernatant) for 2 h. Finally, Atto647-conjugated goat anti-mouse (1:1000) was added for 60 min. The coverslips were mounted onto a microscope slide and analysed by STED microscopy. 

### 4.7. Treatment of Succinimidyl Ester Labelled Mycobacteria with LL-37

First, 1 × 10^9^ Mycobacteria were washed twice with staining buffer (PBS, 0.5% Tween 80 (Roth, Karlsruhe, Germany), 0.2M sodium bicarbonate (Roth, Karlsruhe, Germany); pH of 8.8). Bacteria were then incubated with staining buffer containing the succinimidyl ester Atto647N (1 µg/mL, Sigma-Aldrich, Steinheim, Germany) for 1 h at 37 °C. Afterwards, bacteria were washed three times and resuspended in the appropriate medium. Bacterial viability after Atto647N labelling was ca. 80% in comparison to unlabelled *Mtb*, as determined by ^3^H-Uracil proliferation assay. For STED imaging, labelled bacteria were seeded on a poly-l-lysine coated high precision glass coverslip and exposed to LL-37-TAMRA for 5 min before being fixed with PFA, as detailed previously.

### 4.8. Detection of Microtubule, EEA-1 and LAMP-1 in Macrophages

A total of 5 × 10^6^ Macrophages were infected with *Mtb* (MOI = 50). Infected macrophages were harvested, given into a 12-well plate containing glass coverslips and incubated with LL-37-TAMRA (11 µM) overnight. Cells were fixed with 4% PFA, followed by permeabilization and blocking for 2 h in 0.3% Triton-X and 3% BSA. Cells were incubated with antibodies directed against α-tubulin (rabbit recombinant anti α-tubulin EP1332Y, 1:500; Abcam, Cambridge, United Kingdom) or early endosomes (rabbit polyclonal anti EEA1; 1:500; Sigma-Aldrich, Steinheim, Germany) or lysosomes (rabbit polyclonal anti LAMP1, 1:500; Sigma-Aldrich, Steinheim, Germany) for 2 h at room temperature. Afterwards, cells were washed and incubated with Atto594- or Atto647-conjugated goat anti-rabbit antibodies (1:800; Sigma-Aldrich, Steinheim, Germany) for 2 h. Subsequently, coverslips were mounted onto microscope slides and analysed by STED microscopy. 

### 4.9. STED Setup and Imaging

The STED measurements were performed on a custom build dual-color setup, as described previously [38]. Briefly, a super-continuum laser source (SC-450-PP-HE, Fianium, United Kingdom) with a broad spectral range was used for both, dual color excitation (568 nm and 633 nm) as well as dual color depletion (~720 nm and ~750 nm); beams which were guided onto the back aperture of a high numerical aperture objective (HCX PL APO 100×/1.40–0.70 oil CS, Leica Microsystems, Mannheim, Germany). A piezo stage (733.2DD, Physik Instrumente, Karlsruhe, Germany) was used for scanning the sample and emission was recorded by two avalanche photodiodes (SPCMAQRH-13/14-FC, Perkin-Elmer, Waltham, MA, USA) separately for each channel. The acquisition mode on the same microscope setup could be switched between confocal (diffraction limited resolution, excitation lasers only) or STED (with the depletion beam additionally switched on). Images were first acquired in confocal mode with a pixel size of 100 nm, followed by STED imaging with a pixel size of 20 nm; both with a pixel dwell time of 300 μs. Though a large variety of dye molecules are compatible with the STED technique, the dyes Atto594 and Atto647N were chosen as they have a high fluorescence efficiency and enhanced photostability [54]. Moreover, the fact that both the dyes were chosen from the red-edge of the spectrum resulted in low background from cellular autofluorescence, normally visible in the green and blue spectral bands. 

### 4.10. Flow Cytometry

For investigation of cellular uptake of LL-37, 1 × 10^6^ macrophages were seeded in a 24-well plate and incubated with LL-37-TAMRA (11 µM) for 30 min at 4 °C or room temperature, respectively. Afterwards, cells were harvested using EDTA (1 mM, Sigma-Aldrich, Steinheim, Germany) and transferred into FACS tubes (Falcon, Corning, Corning, NY, USA). Followed by centrifugation at 1300 rpm for 10 min, cells were subsequently washed once with FACS buffer (PBS, 10% sodium azide, 10% FCS). Samples were then measured using a FACS Calibur (BD Biosciences, Franklin Lakes, NJ, USA) and analyzed with FlowJo v10.4.1 (BD, Franklin Lakes, NJ, USA).

### 4.11. Image Processing and Analysis

Image processing was performed using ImageJ software (v 1.52c). All images displayed in this study are processed for brightness/contrast and then displayed with a Gaussian blur of 0.8 to filter out background noise. Images acquired in confocal mode were imaged with a larger pixel size (100 nm) to correlate with the maximum resolution of the confocal technique. The confocal images displayed in this study were resized to 20 nm pixel size in ImageJ, without any filtering, to match the visualization of STED images.

For determining the perimeter of vesicular structures, image analysis was performed in ImageJ using a custom written macro. Raw images were background subtracted (rolling ball method with a radius of 50 pixel) to remove noise and then smoothed. A region of interest (ROI) was manually selected enclosing only one cell (Appendix A). Next, binary masks were created for all EEA1 labelled vesicles by automatic thresholding within the ROI [55]. Perimeters of all detected masks were determined by using the ‘Analyze Particles’ tool in ImageJ. In order to reduce noise, objects with an area smaller 10000 nm^2^ were discarded in accordance with an expected minimal endosomal size of r > 50 nm. 

### 4.12. Statistical Analysis

All statistical analyses as detailed in the legends were performed using GraphPad Prism v8.2.1 (GraphPad Software, La Jolla, CA, USA). A difference in results giving a *p*-value < 0.05 was determined as significant.

## Figures and Tables

**Figure 1 ijms-21-06741-f001:**
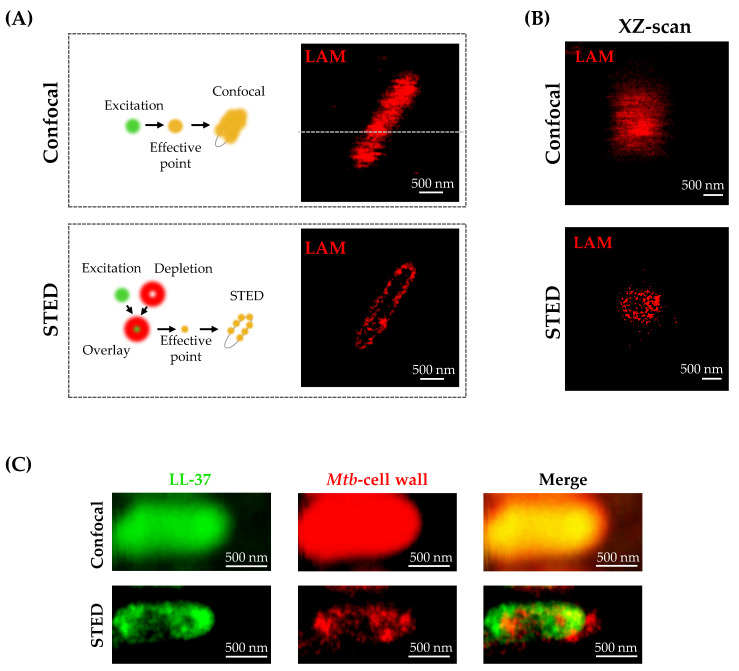
Comparison of Confocal and Stimulated Emission Depletion (STED) Microscopy. (**A**) Schematic visualization of the difference between conventional confocal imaging and STED microscopy (left). Extracellular *Mycobacterium tuberculosis* (*Mtb*) was stained against the cell wall component lipoarabinomannan, followed by imaging of the same bacteria with confocal microscopy (top) and STED microscopy (bottom). Shown is one representative image of ten examined extracellular *Mtb* per independent experiment (n = 5). (**B**) A XZ-scan (along the dotted white line shown in (**A**) of the same lipoarabinomannan (LAM) labelled bacterial cell is shown in confocal and STED mode. Scale bar is 500 nm. (**C**) Dual color experiment of extracellular *Mtb*. Extracellular *Mtb* was labelled with fluorescein succinimidyl ester Atto 647N and incubated with LL-37-TAMRA for 5 min. Image shows LL-37-TAMRA (green), extracellular *Mtb* (red) and merged channels for confocal microscopy (top) and STED microscopy (bottom). Depicted is one representative image of a single extracellular *Mtb* out of ten examined bacteria per independent experiment (n = 3). Images were first acquired in confocal mode, followed by STED imaging with a pixel dwell time of 300 μs and a pixel size of 20 nm.

**Figure 2 ijms-21-06741-f002:**
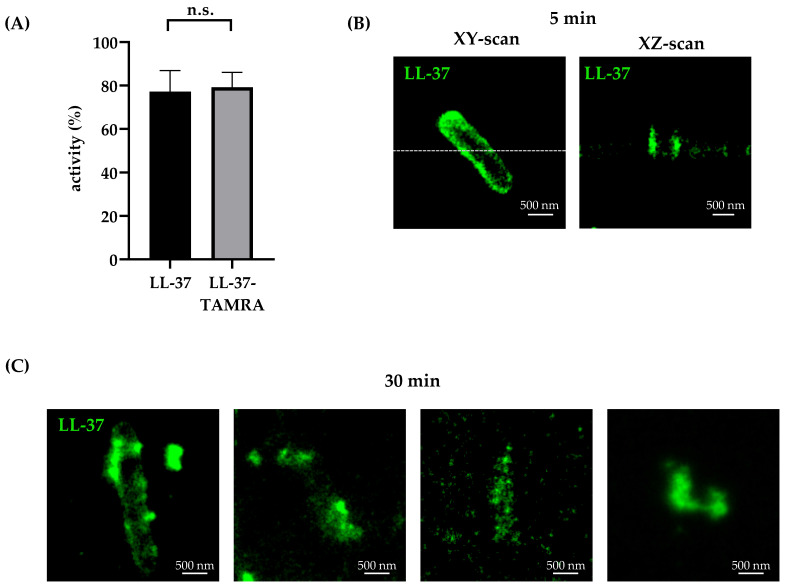
Effect of LL-37-TAMRA on extracellular *Mtb*. (**A**) Antimicrobial activity of unconjugated LL-37 and LL-37-TAMRA (11 µM) against extracellular *Mtb* was determined by incorporation of ^3^H-Uracil. The graph shows the mean percentage activity ± SD of five independent experiments. Statistical analysis was performed using the Kolmogorov–Smirnov-test to compare cumulative distributions (*p* > 0.99). (**B**) Unstained extracellular *Mtb* were treated with LL-37-TAMRA for 5 min. An XZ-scan of the bacteria (along the dotted white line) is shown. (**C**) Unstained extracellular *Mtb* were treated with LL-37-TAMRA for 30 min. Examples of a cell with relatively maintained structural integrity (left) and completely disrupted cellular morphology (right) are shown. (**B**,**C**) Depicted images show one representative *Mtb* out of ten examined bacteria per experiment (n = 3). Images were acquired by STED imaging with a pixel dwell time of 300 μs and a pixel size of 20 nm. Scale bar is 500 nm.

**Figure 3 ijms-21-06741-f003:**
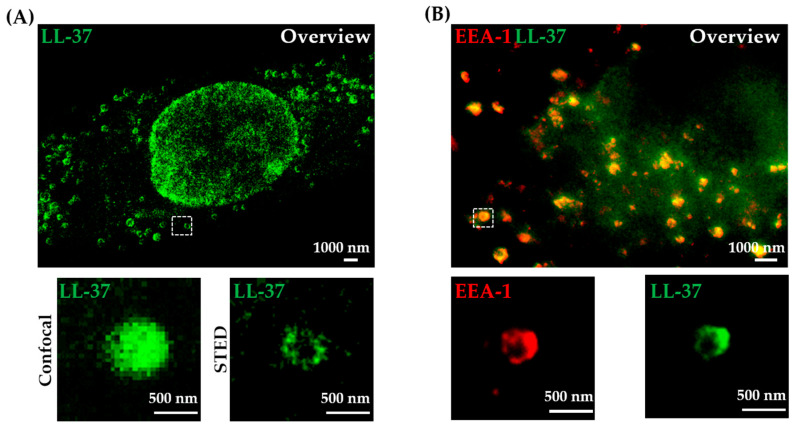
Internalized LL-37-TAMRA localizes with early endosomes in uninfected human macrophages. Cells were incubated with LL-37-TAMRA for 30 min. (**A**) Depicted image shows internalization of LL-37-TAMRA (green) for a representative area of one donor (n = 3). Top image represents an overview of a cell, while bottom images show region of interest for a single vesicle (dashed box magnified approximately five times) with confocal (left) and STED (right) settings. (**B**) Early endosomes were labelled using an anti-EEA1 antibody. Shown is a representative cell of one donor (top, n = 3) with the region of interest for a single vesicle (dashed box magnified approximately five times, bottom). Early endosomes are depicted in red, LL-37-TAMRA in green. Images were first acquired in confocal mode, followed by STED imaging with a pixel dwell time of 300 μs and a pixel size of 20 nm. Scale bar is 1000 nm for overview images and 500 nm for the shown region of interest.

**Figure 4 ijms-21-06741-f004:**
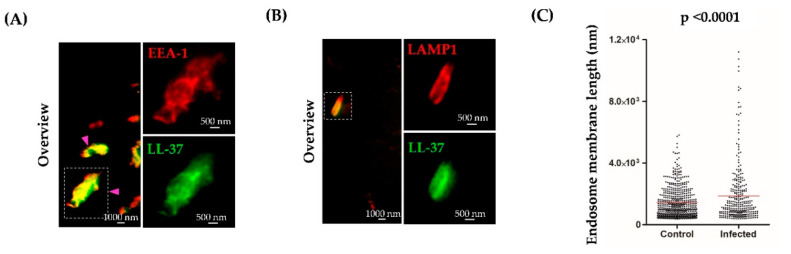
LL-37 colocalizes with early endosomes and phagolysosomes in infected macrophages. Macrophages were infected with *Mtb* (multiplicity of infection; MOI = 50) and incubated with LL-37-TAMRA overnight. Early endosomes or phagolysosomes were labelled using an anti-EEA1 (early endosome antigen 1 protein) or anti-LAMP1 (lysosomal-associated membrane protein 1) antibody, respectively. LL-37-TAMRA is depicted in green and early endosomes or phagolysosomes in red. (**A**,**B**) Images show representative areas of three donors, with specific regions for EEA1 or LAMP1. (**C**) Graph shows the endosome membrane length in uninfected and infected macrophages after LL-37 incubation (n = 4). Statistical analysis was performed using a Students t-test for unpaired samples. Images were acquired using dual color STED microscopy. Scale bar is 1000 nm for overview images and 500 nm for the shown region of interest.

**Figure 5 ijms-21-06741-f005:**
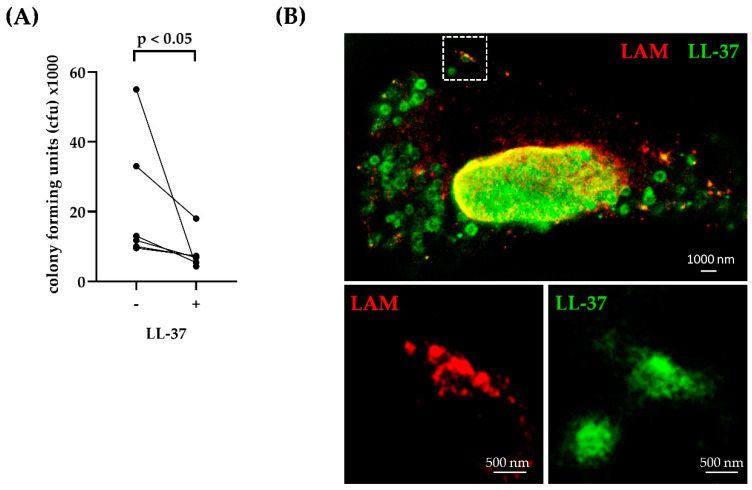
Antimicrobial effect of LL-37-TAMRA against intracellular *Mtb*. (**A**) Macrophages were infected with *Mtb* (MOI = 5) for 2 h, followed by incubation with LL-37-TAMRA for 72 h. Afterwards, cells were lysed and plated on 7H11 agar plates to determine mycobacterial growth via colony forming units (CFU). The graph gives the individual results of all six donors investigated. Statistical analysis was performed using a Wilcoxon matched-pairs signed rank test (*p* < 0.05). (**B**) Macrophages were infected with unstained *Mtb* and incubated with LL-37-TAMRA overnight. Internalized *Mtb* was labelled using an anti-LAM-antibody. Representative area of three different donors is shown. *Mtb* are depicted via lipoarabinomannan (LAM) in red, LL-37-TAMRA in green. Images were acquired using dual color STED microscopy. Scale bar is 1000 nm for overview images and 500 nm for the shown region of interest.

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
