# Peer review of "Super-Resolution Microscopy Reveals a Direct Interaction of Intracellular Mycobacterium tuberculosis with the Antimicrobial Peptide LL-37"

_ijms, 2020, doi:10.3390/ijms21186741_

Round 1
Reviewer 1 Report
This manuscript describes use of the Super-resolution technique STED to perform in-situ measurements of the effect of an AMP in mycobacteria inside macrophages. As far as I am aware, this is the first time such experiments have been performed in such a challenging environment.
This Manuscript is extremely clear, concise and well written, and I believe the subject timely.
I have three small criticisms.
1. On page 9, lines 254 to 259. The authors explain the advantages of STED technique over “normal” confocal microscopy:, They state: “The high spatial resolution of the STED technique thus 256 demonstrated the colocalization of Mtb and LL-37 macrophage compartments, 257 without the pitfalls of ‘false co-localization’ seen in diffraction-limited optical 258 microcopy.”
The authors should say something like “Greatly reducing the problem of co-localisation”, instead of “Without”, as the ultimate resolution of the STED (20nm), is still much larger than the size of the fluorophores in use. Thus, there still is (an always will be), some compromise due to the resolution of the technique.
2. It is of paramount importance in this paper, that the imaging conditions are described well. The way the paper is written, it is not fully clear how the confocal imaging compared to the STED conditions:
“Acquisition mode on the same microscope setup could be switched between confocal (diffraction limited resolution, excitation lasers only) or STED (with the depletion beam additionally switched on). Images were first acquired in confocal mode, followed by STED imaging with a pixel dwell time of 300 μs and a pixel size of 20 nm.”
Can the authors please make it clear if the Dwell time and pixel size were the same for both techniques, or if they differed?
3. I would further suggest that the author could compare their results to those found in the following papers:
Fluorescence Imaging of Bacterial Killing by Antimicrobial Peptide Dendrimer G3KL
By:Gan, et al., ACS INFECTIOUS DISEASES Volume: 5 Issue: 12 Pages: 2164-2173
and
Super-resolution Stimulated Emission Depletion-Fluorescence Correlation Spectroscopy Reveals Nanoscale Membrane Reorganization Induced by Pore-Forming Proteins
By:Sarangi, et al, Langmuir V 32 Issue: 37 Pages: 9649-9657
Reviewer 2 Report
Recommendation: major revision
The manuscript submitted by Deshpande and colleagues aims to assess the super-resolution technique of stimulated emission depletion (STED) microscopy to study the intracellular localization of antimicrobial peptides (AMP) such as LL-37. In this study the authors propose the use of STED microscopy as a tool for directly observing the interaction of AMP with the intracellular bacteria Mycobacterium tuberculosis inside primary human macrophages.
The study compares the images obtained by conventional confocal microscopy and STED microscopy, showing better image resolution for the latest technique.
I think this is a very interesting paper. Most results support the conclusions but more image acquisition with STED microscopy is necessary for a complete contrast with conventional confocal microscopy and some improvements in the methodology are also needed
Positive feedback
- Due to the size of the peptide (37aa) and taking into account that it is bound to TAMRA, it will not pass through the mycobacterial membrane without first disrupting it, which the researchers were able to determine.
- The authors present the correct targets for microscopic analysis and it is quite favorable that they do the count of CFU's after each treatment, including the study in infected macrophages.
- Usually they work with MOI 5 or 10, so the design of the experiment was fine.
- It is very interesting that they show this technique, because as they comment, confocal microscopy has too much noise and not enough resolution compared to STED microscopy.
.
Materials and Methods
This item must be improved:
In lines 348, 390 and 408: The number of cells (macrophages) used for the infection is missing.
In lines 331 and 359: In sonication, details such as the watts and pulses used are missing.
In the Quantification of intracellular mycobacterial growth, 2x105 macrófagos infectados (MOI:5) fueron incubados con LL-37 a 11uM por 4 días but in Imaging of the LL-37 and Mtb in macrophages and Detection of EEA-1 and LAMP-1 in macrophages were used a MOI:50, Why?
The methodology shows changes in the incubation times of macrophages infected with LL-37 peptide; for example, In lines 392 and 410, Why do they use different times?
In the 3H-Uracil proliferation assay, 2x106 bacteria were used which were incubated with LL-37 (the concentration of peptide used is missing) and in Treatment of extracellular Mtb with LL-37-TAMRA, 20x106 bacteria (replace by 2x107 ) and a lower amount of peptide (22nM) were used. Why in this test a lower peptide concentration was used than in the other tests? It is 500 times less peptide than the other assays, why this difference?
In line 400 The number of bacteria used is missing
In flow cytometry test, a smaller number of macrophages were used and incubated with the LL37 peptide without indicating its concentration and this time the incubation was performed for 30 min. Again why?
I have the impression that the amount of cells, the concentration of peptide to be used and the incubation times need to be standardized? Please show the standardization of these conditions that clearly support the concentrations, times and cells used.
Results
- LAM DETECTION
In this first part of the results, the detection of LAM is shown, but this is not detailed in materials and methods; it is suggested to add it.
Figure 1A and 1B: To corroborate what is presented in the selected image, a quantification of the fluorescence observed by STED in the 10 images analyzed by replication showing a graph with SD is requested.
In line 112-113 it is indicated that the incubation of the Mtb bacteria with the peptide was done for 5min, why is this indicated if in materials and methods the incubation of the peptide with the bacteria is indicated that it was done for 3 days? Furthermore, it is not detailed in methods.
- INTERACTION BETWEEN Mtb AND ANTIMICROBIAL PEPTIDES
The authors indicate in Fig 1B that ... 'In the STED image, clusters of LL-37 are clearly distinguishable from them labelling of the cell wall'...but when looking at the image both individual and aggregate peptide markings are observed, so if the mode of action of LL-37 in the bacterium is to be determined, it is not possible to determine it here.
Like I said before, I agree that the STED, in terms of resolution, is much better than the confocal but when I look at FIG 1B, it seems that the peptide surrounds the bacteria and it is not clear if it is interacting with the cell wall of the bacteria or has some intracellular target... I suggest to indicate in the text that there is a type of collocation and with arrows indicate where this collocation is seen. To clarify this, is STED microscopy compatible with Z-Stack acquisition mode? Can the authors include some images with this approach?
- EFFECT OF LL-37 ON EXTRACELLULAR Mtb
In Line 136, it was indicate that the bacteria were incubated for 5 and 30 min with the peptide and in method section said 3 days…why?
The authors indicate that ... 'After 30 min the cell wall appeared disrupted with few discernible peptide clusters. How do you know that the rupture of the cell wall bacterial occurred? Where do you see this? When you look at the morphology of the bacteria you don't see an evident change between 5 and 30 minutes. Is it possible that the peptide has internalized and therefore decreases the mark on the surface? The authors indicate that this does not happen, but if this equipment allows to make a Z- Stack, it would be good to do it to check what is indicated.
Also, a quantification of the fluorescence must be done to see changes observed at 5 and 30 min.
On the other hand, has it been demonstrated through another technique that the peptide cluster observed is the way in which this peptide acts?
Finally, if the authors suggest that there is a disruption in the cell wall, sequential images in time to observe this change should be included ( for example at 5, 15, 30, 45 and 60 min) and also a graph of quantification of cell wall and cytosol fluorescence.
- LL-37 IS TAKEN UP BY MACROPHAGES AND COLOCALIZES WITH EARLY ENDOSOMES
In figure 3, a picture with the complete macrophage showing the bacteria inside the cell, i.e nucleus- cytoskeleton cell staining is required.
Indicate magnification used in Figure 3B and 3B.
- LL-37 LOCALIZES IN ENDOSOMES AND LYSOSOMES OF INFECTED MACROPHAGES
The authors in the introduction indicate that …‘Mtb prevents the formation of acidic phagolysosomes and the bacteria sequester in the phagosomes protected from degradation’, so why the formation of a phagolysosome was observed in infected macrophages? It is possible that the peptide may activate the maturation process from the endosome to the lysosome, as indicated by the authors in their discussion. However, a picture of infected macrophages without the peptide is requested to see if phagolysosome formation occurs.
